# Removal of Zinc from Aqueous Solutions Using Lamellar Double Hydroxide Materials Impregnated with Cyanex 272: Characterization and Sorption Studies

**DOI:** 10.3390/molecules25061263

**Published:** 2020-03-11

**Authors:** Nacera Boudaoud, Hafida Miloudi, Djamila Bouazza, Mehdi Adjdir, Abdelkader Tayeb, Agustin Fortuny, Hary Demey, Ana Maria Sastre

**Affiliations:** 1Laboratory of Chemistry of Materials, University of Oran1, B.P 1524 El M’naouer-Oran, Algeria; boudaoud_2007@yahoo.fr (N.B.); bouaza_dj@yahoo.fr (D.B.); m_adjdir@yahoo.fr (M.A.); tayeb_aek@yahoo.fr (A.T.); 2Department of Chemical Engineering, Universitat Politècnica de Catalunya, ETSEIB, Diagonal 647, 08028 Barcelona, Spain; agustin.fortuny@upc.edu (A.F.); ana.maria.sastre@upc.edu (A.M.S.); 3Faculty of technology, Department of engineering process, University of Saïda Dr. Tahar Moulay, 20000 Saida, Algeria

**Keywords:** cyanex 272, impregnation, layered double hydroxides (LDH), sorption, zinc

## Abstract

Removal of heavy metals from wastewater is mandatory in order to avoid water pollution of natural reservoirs. In the present study, layered double hydroxide (LDH) materials were evaluated for removal of zinc from aqueous solutions. Materials thus prepared were impregnated with cyanex 272 using the dry method. These materials were characterized through X-ray diffraction (XRD), Fourier transform infrared (FTIR), and thermal analysis. Batch shaking adsorption experiments were performed in order to examine contact time and extraction capacity in the removal process. Results showed that the equilibrium time of Zn (II) extraction is about 4 h for Mg_2_Al-CO_3_ and Mg_2_Al-CO_3_-cyanex 272, 6 h for Zn_2_Al-CO_3_, and 24 h for Zn_2_Al-CO_3_-cyanex 272. The experimental equilibrium data were tested for Langmuir, and Freundlich isotherm models. Correlation coefficients indicate that experimental results are in a good agreement with Langmuir’s model for zinc ions. Pseudo-first, second-order, Elovich, and intraparticular kinetic models were used to describe kinetic data. It was determined that removal of Zn^2+^ was well-fitted by a second-order reaction kinetic. A maximum capacity of 280 mg/g was obtained by Zn_2_Al-CO_3_-cyanex 272.

## 1. Introduction

Heavy metal pollution is a widespread environmental problem that can pose serious threats to human health and the ecosystem [1,2]. Heavy metal ions are stable and persistent environmental contaminants, since they cannot be degraded or destroyed. The remediation of heavy-metal-contaminated soil and water has always been a hot topic of environmental science and technology. Many technologies have been established for the treatment of wastewater containing heavy metal [2,3,4,5,6].

Layered double hydroxides (LDH) or anionic clays are lamellar ionic compounds, containing a positively charged layer and exchangeable anions in the interlayer [2,7]. They consist of brucite-like layers, with a partial M^II^ for M^III^ substitution, leading to an excess of positive charge compensated with anions (which represent an important class of ionic lamellar solids). Layered double hydroxides are usually represented by the general formula [M^2+^_1−x_M^3+^_x_ (OH)_2_ (A^n−)^_x/n_]·yH_2_O [8,9,10,11].

LDHs have been studied for their potential use in a wide range of important areas, including catalysis [12,13] and biomedical science [14,15]. They have been applied as catalysts and as effective supports for immobilization of enzymes and noble metal catalysts [16]. The structure of these solids is derived from the brucite structure in which trivalent cations partially substitute the divalent ones. This substitution gives rise to positively charged layers balanced with interlayer anions; water molecules also exist in the interlamellar region [7,17]. The divalent (Mg^2+^, Zn^2+^, Cu^2+^, etc.) and trivalent (Al^3+^, Cr^3+^, Fe^3+^) cations occupy the center of [M^2+^/M^3+^] (OH)_6_ octahedral units, and A_n_^−^ is an inorganic or organic anion.

The calcination of LDH_s_ at 550 °C transforms them into mixed oxides (Mg-Al-550), which give the layered double hydroxide after rehydration in the presence of the desired anion. This phenomenon is known as the memory effect [18]. LDHs can be found in nature as minerals such as hydrotalcite (Mg-Al LDH), pyroaurite (Mg-Fe LDH), and takovite (Ni-Al LDH), whose interlayer anion is carbonate (most of the time), although chloride and sulfate are present sometimes. Layered double hydroxides are also synthesized quite easily at laboratory scale.

In natural environments, LDH minerals are significant to determine the uptake of heavy metal ions [19]. Divalent cations can be adsorbed by aluminum oxides and may form hydrotalcite-like minerals, such as Ni-Al LDH [20] and Zn-Al LDHs [21]. This process can obviously reduce heavy metal concentration in aquifers, and subsequent migration and bioavailability. In the past few years, many reports on adsorption of various contaminants on LDHs have been published. The contaminants include oxyanions, monoatomic anions, organic compounds, and gas. Only a few studies have focused on sorption of cations on LDHs. An important pollutant to be considered is zinc, which is highly present in industrial effluents (such as metallurgical and ceramic wastewaters). The introduction of waters contaminated with zinc into the ecosystems represents a serious environmental concern today.

Recently, some authors reported LDHs containing different chelating agents [22,23,24,25,26,27,28,29] as well as the metal cation uptake achieved by these materials. Higher retention zinc capacities were obtained after impregnation of bentonite by cyanex 272 [30]. Referring to the literature, no studies on LDH functionalized with organophosphorus acids as a ligand for the trapping of metals have been cited. This work deals with the use of impregnated materials by cyanex 272 as an adsorbent of Zn (II). Extraction experiments at different conditions were performed. The sorption capacities, isotherms, kinetics, and sorption mechanisms were determined and discussed in the next sections.

## 2. Experimental

### 2.1. Materials

Stock solutions of Zn (II) (1 g/L) were prepared by dissolving Zn(NO_3_)_2._6H_2_O in deionized water. The salts and reagents used in this work (such as Al(NO_3_)_3_.9H_2_O (Biochem, 99,99%), Mg(NO_3_)_2._6H_2_O (Biochem, 99.9%), Na_2_CO_3_ (99%), NaOH (Chemopharma, 98.8%), NaNO_3 _(Sigma Aldrich), HNO_3_, and ethanol) were provided by Panreac (Barcelona, Spain). The LDH materials were synthesized by co-precipitation, and Di (2,2,4- trimethylpentyl) phosphinic acid was provided by Cyanamid (New Jersey, USA) as cyanex 272 (85%) (Figure 1).

### 2.2. Preparation of Sorbents

The LDH materials were prepared according to the coprecipitation method [26], with a molar ratio metal/aluminum 2:1 (i.e., Zn/Al and Mg/Al), at a constant pH of 10 (which was monitored by a combined glass electrode connected to a pH meter Jenway-3310). The synthesis consists in carrying out the controlled precipitation of a solution containing the two nitrate salts, Mg (NO_3_)_2._6H_2_O (0.5 M) and Al(NO_3_)_3_.6H_2_O (0.25 M), by progressive addition of a basic solution containing 1 M of NaOH and 2 M of Na_2_CO_3_, with magnetic stirring at room temperature. The pH is kept constant at 9.0 ± 0.1 by adding a 2 M sodium hydroxide solution. The suspension was stirred for 24 h at 80 °C for maturation and then centrifuged and separated. The solid was rinsed three times with abundant distilled water and dried at 80 °C.

### 2.3. Preparation of Impregnated Sorbents

The cyanex 272 impregnated LDH was prepared in accordance with the dry impregnation method [31,32]. Cyanex 272 was dissolved in ethanol, and then the solution was left in contact with the LDH material under magnetic stirring at room temperature until total evaporation of ethanol. The obtained solid was washed with HNO_3_ solution (0.1 M) in order to avoid the dissolution of cyanex 272 in the aqueous phase. Finally, this solid was dried under atmospheric pressure at 80 °C for 24 h.

### 2.4. Characterization of Sorbents

The materials were characterized using a Fourrier transform infrared spectrophotometer (FTIR) on the pelletized solids in between (4000–400 cm^−1^), on a Bruker Alpha apparatus, and X-ray diffraction (XRD) powder patterns were obtained with CuKα_1_ radiation (1.5406 Å) on a powder diffractometer SIEMENS D501. Thermal–gravimetric analyses (TGA) of the synthesized materials were conducted under nitrogen medium from room temperature to 800 °C with a temperature rate of 5 °C/min, on a Mac Science 2000S type instrument.

### 2.5. Methods

#### 2.5.1. Metal Sorption Procedure

The removal of Zn (II) from aqueous solutions was carried out through batch experiments at 25 °C. A fixed amount of sorbent (0.1 g) was mechanically mixed in polypropylene tubes with 10 mL of aqueous metal solution at a known initial metal concentration (and fixed initial pH 1), for 48 h of agitation time to achieve the equilibrium. Then, the solid phase was separated from the aqueous phase through centrifugation (at a speed of 8000 rpm); aliquots of five milliliters were withdrawn for analysis using the atomic absorption technique (AAS) on a Perkin-Elmer 2380 spectrophotometer. The experiments were performed in duplicate, and the standard deviation was estimated in the order of ±2%.

#### 2.5.2. Effect of Contact Time

The experimental protocol was performed following the procedure described in Section 2.5.1. This means that several recipients containing 0.1 g of solid sorbent were prepared. Then, 10 mL of the mother solution was added in each one (at fixed initial pH and initial metal concentration). The recipients correspond to each sampling time (contact times ranging from 0 to 48 h); these were mixed with the sorbent and agitated at the same started time (t_0_). The agitation speed of the recipients was stopped according to the corresponding contact time, and the aqueous phase was immediately separated from the solid phase.

In this way, the variation of volume due to the aliquot sampling is avoided (the same contact volume is guaranteed for each sampling time). The experiments were performed twice for each sorbent material, and the standard deviation was ±2%. The pH was also monitored, and the obtained data were fitted with the pseudo-first order, pseudo-second order, Elovich, and intraparticular diffusion models.

#### 2.5.3. Equilibrium Studies

The sorption capacity of Zn (II) by the different materials was carried out as described in Section 2.5.1 at initial concentrations ranging from 100 to 900 mg/L. The pH was monitored, and the samples were collected after 24 h (enough time to achieve equilibrium). In order to understand the adsorption mechanism, different isotherm models were employed for fitting the experimental data (Langmuir and Freundlich models).

## 3. Results and Discussion

### 3.1. Characterization of the Materials

#### 3.1.1. X-Ray Diffraction Analyses (XRD)

The XRD patterns of Mg_2_Al-CO_3_, Zn_2_-Al-CO_3_, and Mg_2_Al-CO_3_-cyanex 272 LDHs are given in Figure 2. The diffraction peaks are at low angles, as indexed with (003) and (110) for a rhombohedral symmetry (3R1). It is known that 3R1 is featured by the diffraction peaks of (012), (015), (018), etc. The interlayer spacing (0.307 nm) is in good agreement with that reported by Carriazo et al. [33], who found a value of 0.304 nm. The substitution of Mg by Zn does not affect the basal spacing of the obtained LDH, which can be attributed to their very close ionic radii of 72 and 74 pm, respectively. However, the use of nitrate ion instead of carbonate ion enhanced the basal spacing of LDH from 0.784 to 0.840 nm, which may be related to their bond length. It is noted that the bond length CO in carbonate ion is around 0.128 nm, whereas the bond length NO in nitrate ion is circa 0.132 nm.

The interlayer spacing d003 of LDHs was calculated from those of the (003) by the given relation *c/3* = d003 [34,35]. The cell parameter a, which is related to the metal–metal interatomic distance within the sheets, is calculated from (110) under the following relation *a/2* = d110 [34,36,37,38]. These patterns also indicate that the intercalation of cyanex 272 in Mg_2_Al-CO_3_ and Zn_2_Al-CO_3_ gives rise to an increase in basal spacing from *d* = 7.702 Å to *d* = 15.99 Å and from 7.839 to 16.65 Å, respectively. The unit cell parameters of the synthesized LDHs are summarized in Table 1.

#### 3.1.2. Infrared Spectroscopy (FTIR)

FTIR spectra of the different materials and cyanex 272 are presented in Figure 3. The strong peaks at 3585, 3491, 3462, and 3286 cm^−1^ correspond to Zn_2-_Al-CO_3_, Mg_2_Al-CO_3_, Zn_2_Al-CO_3_-cyanex 272, and Mg_2-_Al-CO_3_-cyanex 272, respectively, and are attributed to the stretching vibration of hydroxyl groups associated with the interlayer water molecules and hydrogen bonding [39,40,41]. The intensity of the -OH bands depends strongly on the hydration rate, the density charge of the sheets, and the nature of the cation. The stretching vibration of CO_3_^2−^ in the LDH interlayer is localized in the region of 1350. For M^II^/AlLDH, bands in the lower energy region correspond to the lattice vibration mode, such as the translation vibrations of M^II^-OH at 750 cm^−1^, and deformation vibration of OH-M^II^-Al-OH at 575 cm^−1^ was observed [10] (Figure 3). The presence of cyanex 272 in the impregnated materials was proven by the appearance of new stretching vibration bands of aliphatic C-H of the alkyl chain CH_2_ and CH_3_ at 2960 and 2860 cm^−1^, respectively, in the corresponding impregnated LDH spectra.

#### 3.1.3. Thermal–Gravimetric Analyses (TGA)

The different steps of both the Mg_2_Al-CO_3_ and Zn_2_Al-CO_3 _LDH decomposition are shown in Figure 4. The TGA patterns are characterized by a weight loss of the interlayer water in the temperature range of 50–250 °C around 20% and 5.5% corresponding to Mg_2-_Al-CO_3_ and Zn_2_Al-CO_3_, respectively. This difference can be attributed to the loss of both water adsorbed on external surfaces and more strongly held interlayer water. Based on the literature [11], the mass loss around 290 °C could be attributed to the Al–OH dihydroxylation, while the Mg–OH dehydroxylation and decarbonation occurs simultaneously at 300 °C. The dehydroxylation and decarbonation steps occur between approximately 300 and 850 °C, which lead to the formation of metal oxide into a spinel phase [42,43,44].

### 3.2. Sorption Studies

#### 3.2.1. Effect of contact time

The study of the effect of contact time on the extraction of zinc by the layered double hydroxide materials was carried out for variable contact times of up to 48 h. The experimental results are shown in Figure 5. It is worth noting that the equilibrium was reached after 4 h for Mg_2_Al-CO_3_ and Mg_2_Al-CO_3_-cyanex 272 materials, 6 h for Zn_2_Al-CO_3_, and 24 h for Zn_2_Al-CO_3_-cyanex 272 materials. The results obtained show that the extraction time of Zn (II) varies according to the sorbent. This process is characterized by a relatively slow equilibrium time. The most effective materials were Zn_2_Al-CO_3_ and Zn_2_Al-CO_3_-cyanex 272. The pH of the solution is a very important parameter to take into consideration in sorption processes. The sorbent materials were evaluated in this work at initial pH_0_ 1 in order to simulate the acid conditions of the real metallurgical effluents (in which zinc pollution is frequently present), and also to avoid metal precipitation due to the pH increases.

Appendix A reports the pH variation using the impregnated and non-impregnated LDH materials. As expected, after contact with the sorbents, the equilibrium pH of the effluents increases for two main reasons: (i) the competition between protons and metals ions for the active sites; (ii) probably, an insufficient washing procedure (during the LDH manufacturing process), which could cause the release of trace amounts of NaOH on aqueous applications.

#### 3.2.2. Fitting of the Kinetic Data

This study was carried out to determin the kinetic order of the zinc extraction process by the studied materials; different kinetic models were applied to the experimental results. It is worth noting that the fitting of the data does not mean that the principles of the models are verified, but it helps to understand the involved mechanisms. Inglezakis et al. [45] pointed out that the sorption mechanism cannot be directly assigned by simply fitting the kinetics equations; the knowledge should be supported by combining the analytical surfaces techniques (e.g., FTIR, XRD, SEM).


**a. Model of pseudo-first order**


For a pseudo-first order reaction, the rate law is expressed as [46]:(1)$$\frac{dq_{t}}{dt}=k_{1}(q_{e}-q_{t})$$
where *k*_1_: the rate constant (min^−1^), *q_e_*: amount of the metal ion extracted at equilibrium per unit of sorbent (mg/g), and *q_t_*: amount of the metal ion extracted at time *t* per unit of sorbent (mg/g).

After Equation (1) is integrated between 0 and t for the time and between 0 and *q_t_* for the quantity of the metal ion extracted, we obtain:(2)$$Ln(q_{e}-q_{t})=Lnq_{e}-k_{1}.t$$

The results are shown in Figure 6.


**b. Model of pseudo-second order**


For a pseudo-second order reaction, the rate law is expressed as [47,48,49,50,51]:(3)$$\frac{dq_{t}}{dt}=k_{2}.{\left(q_{e}-q_{t}\right)}^{2}$$
where *k*_2_ is the rate constant (g.mg^−1^h^−1^). After Equation (3) is integrated, the following integrated law is obtained:(4)$$\frac{1}{q_{e}-q_{t}}=\frac{1}{q_{e}}+k_{2}.t$$

Equation (4) can be rearranged as follows:(5)$$\frac{t}{q_{t}}=\frac{1}{k_{2}.q_{e}^{2}}+\frac{1}{q_{e}}t$$

The results are shown in Figure 7; the rate constants and correlation coefficients were determined by plotting Ln (*q_e_* − *q_t_*) as a function of time *t* for pseudo-first order and *t*/*q_t_* versus time for pseudo-second order.

After comparison of the correlation coefficients for zinc ion, it is noted that the experimental points were consistent with the pseudo-second order model. Based on this model, the correlation coefficients and amounts of metal extracted at equilibrium are determined after recalculation by taking into account all the experimental points (Table 2). The obtained results are in good agreement with those determined experimentally, and these also agree with those found by Azizian [52], in which it was found that the pseudo-second order model is more suited when the initial metal concentration is low. Ho and McKay [48] stated that for all investigated systems, the chemical reaction seems significant in the rate-controlling step, and the pseudo-second order chemical reaction kinetics provide the best correlation of the experimental data.


**c. Model of Elovich**


The Elovich model was applied to the experimental results [53].
(6)$$q_{t}=\beta \ln\left(\alpha \beta \right)+\beta \ln t$$
where *α* is the initial adsorption constant (mg/g.h) and *β* is the desorption constant (g/mg). The results according to the Elovich model are shown in Figure 8.


**d. Model of intraparticular diffusion**


For the intraparticular diffusion model, the equation is expressed as [54]:(7)$$q_{t}=k_{id}\times \sqrt{t}+C$$
where *q_t_* is the amount of Zn(II) adsorbed at time *t*, *C* is the intercept, and *k_id_* (mg.g^−1^.h^−0.5^) is the intraparticular diffusion rate constant. This latter is determined from the linear plot presented in Figure 9 of q_t_ versus t ^0.5^, and it is usually used to compare the mass transfer rates. If the plot does not pass through the origin, this is indicative of some degree of boundary layer control, and intraparticle diffusion is not the sole rate-limiting step. The intercept values are 1.311, 3.279, 1.962, 1.541, and 3.690 mg.g^−1^ for Mg_2_Al-CO_3_, Mg_2_Al-CO_3_-cyanex 272, Zn_2_Al-CO_3_, and Zn_2_Al-CO_3_-cyanex 272, respectively (Table 2), which indicates that pore diffusion is not a rate-limiting step [51,54,55,56,57].

Comparing the different models studied, we notice that the pseudo-second order model is the best adapted to the experimental results, which confirms a better zinc retention; this does not exclude a 2nd order chemisorption in any case.

#### 3.2.3. Equilibrium Studies

The sorption capacity (q)_e_ of Zn (II) by the different materials was carried out as described in Section 2.5.3 at concentrations ranging from 100 to 900 mg/L. It was determined by measuring the amount of metal ion present in solution before (C_i_) and after saturation (C_s_):(8)$$q_{e}=\frac{(C_{i}-C_{e})\times V}{M}$$
where q_e_ is the amount of zinc extracted (mg/g) by the materials, C_i_ is the initial concentration of the solute (mg/L), C_e_ is the concentration of the supernatant after extraction (mg/L), V is the volume of the solution (L), and M is the mass of the adsorbent (g).

The adsorption isotherms studied are shown in Figure 10. In this study, only materials that have shown better zinc retention are retained. The Zn (II) extraction isotherms by Zn_2_Al-CO_3_ and Zn_2_Al-CO_3_-cyanex 272 are of type C (according to the Giles classification); there is in this case a linear relationship between the amount of adsorbed solute q_e_ and the concentration at equilibrium C_e_. This type of isotherm reflects cooperative interactions between the adsorbate and the adsorbent [58].

On the other hand, linear isotherms suggest that the sorption phenomenon is controlled by a sharing process. Pollutant sorption on clays and modified clays is governed by several mechanisms, including solute sharing between the aqueous phase and the solid phase, solvation, and adsorption on the surfaces of the adsorbent [59]. The saturation plateau does not appear; there is a linear increase in the amount adsorbed as a function of the concentration of Zn (II). These results are identical to those obtained by Inacio et al. [36]. The linear adsorption encountered during the fixation of Zn (II) on the materials can be described by the following equation [60]:q_s_ = K_d_.C_e_(9)
where q_s_ is the amount of zinc fixed on the material (mg/g), C_e_ is the concentration of the remaining zinc in the solution after extraction (mg/L), and K_d_ is the adsorption constant. After plotting q_s_ against C_e_, we determined the K_d _values for the different materials (Table 3).

#### 3.2.4. Fitting of the isotherm data

Data from the Zn (II) extraction on the Zn_2_Al-CO_3_ and Zn_2_Al-CO_3_-cyanex 272 materials are modeled according to the following linearized Langmuir and Freundlich equations.

Freundlich equation:(10)$$q_{e}=k_{f}C_{e}^{n}$$
where *k_f_* represents the adsorption capacity and n the intensity of the adsorption. Values of n indicate the type of the isotherm: irreversible (n = 0), favorable (0 < n < 1), and unfavorable (n > 1) [61]. The preceding equation can be given in linear form [62] as:(11)$$\log q_{e}=\log k_{f}+(n)\log C_{e}$$

Langmuir equation:(12)$$q_{e}=\frac{q_{\max}K_{1}C_{e}}{1+K_{1}C_{e}}$$
where *K_l_* is the equilibrium adsorption coefficient (L/mg), *q*_max_ the maximum adsorption capacity (mg/g), *C_e_* the equilibrium solution concentration (mg/L), and *q_e_* is the amount adsorbed at equilibrium (mg/g). The linear form of the Langmuir equation can be written as follows:(13)$$\frac{C_{e}}{q_{e}}=\frac{1}{K_{1}q_{max}}+\frac{C_{e}}{q_{\max}}$$

Plots of Freundlich and Langmuir isotherms of Zn (II) extraction by the different materials are shown in Figure 11 and Figure 12, respectively. Langmuir and Freundlich constants are summarized in Table 3. Examination of the isothermal modeling results of Zn (II) extraction by the different materials and their comparisons with experimental data show that the Langmuir model is better-adapted to the results. The maximum sorption capacities were found as 268.82 and 67.17 mg/g, for Zn_2_Al-CO_3_-cyanex 272 and Zn_2_Al-CO_3_ materials, respectively.

## 4. Conclusions

In this work, the lamellar-double-hydroxide materials impregnated with cyanex 272 were demonstrated to be promising for zinc removal from wastewaters. The experimental results allowed verifying three important issues:

1. The interlayer spacing increases after impregnation of solids, which is relevant for understanding the removal mechanism of metal ions from aqueous effluents.

2. The contact time for achieving the equilibrium was faster for Mg-based materials, which is indicative of the higher diffusion properties for future industrial applications: 4 h for Mg2Al-CO3 and Mg2Al-CO3-cyanex 272, 6 h for Zn2Al-CO3, and 24 h for Zn2Al-CO3-cyanex 272. The pseudo-second-order model fitted better the kinetic data.

3. After impregnation of the materials with cyanex 272, the sorption capacity towards zinc ions is four times larger than that before impregnation (especially for Zn-based sorbents). The Langmuir equation is the most suitable model for fitting the experimental data.

In order to approach real industrial conditions, further experiments will be performed with complex systems to evaluate the influence of multiple metal ions on the sorption capacity. Additionally, the upscaling of the manufacturing production will be studied.

## Figures and Tables

**Figure 1 molecules-25-01263-f001:**
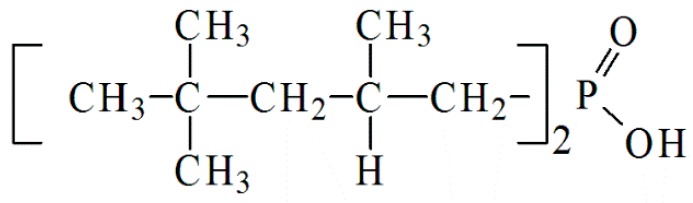
Structure of cyanex 272.

**Figure 2 molecules-25-01263-f002:**
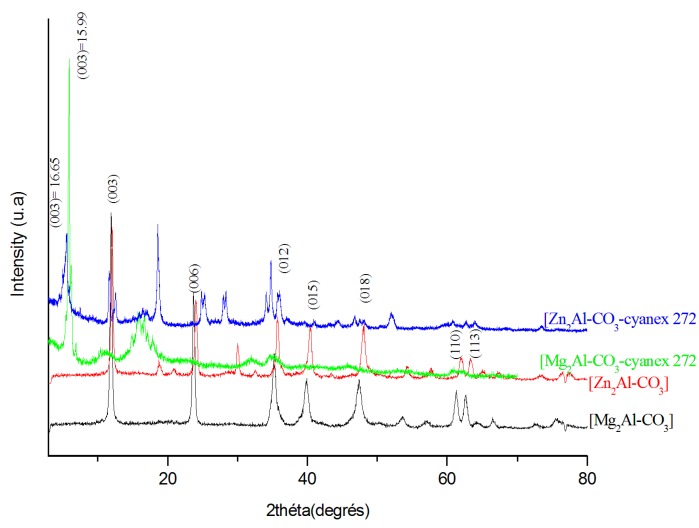
XRD patterns of Mg_2-_Al-CO_3_, Zn_2-_Al-CO_3_, Zn_2_Al-CO_3_-cyanex 272, and Mg_2_Al-CO_3_-cyanex 272.

**Figure 3 molecules-25-01263-f003:**
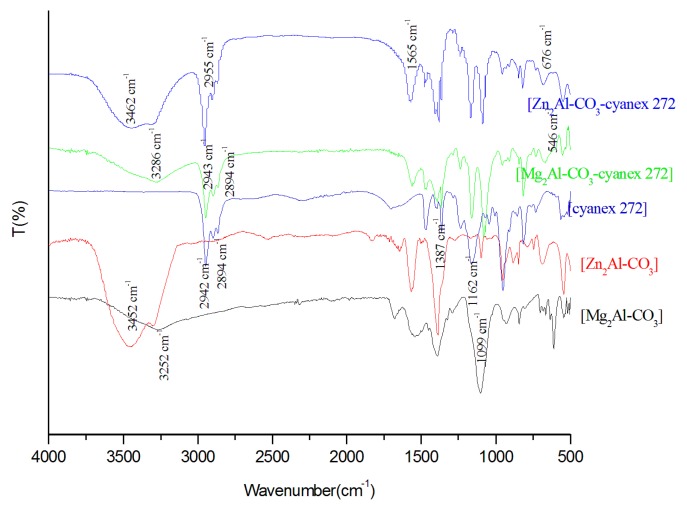
FTIR spectrum of Mg_2-_Al-CO_3_, Zn_2-_Al-CO_3_, Mg_2_Al-CO_3_-cyanex 272, Zn_2_Al-CO_3_-cyanex 272, and cyanex 272.

**Figure 4 molecules-25-01263-f004:**
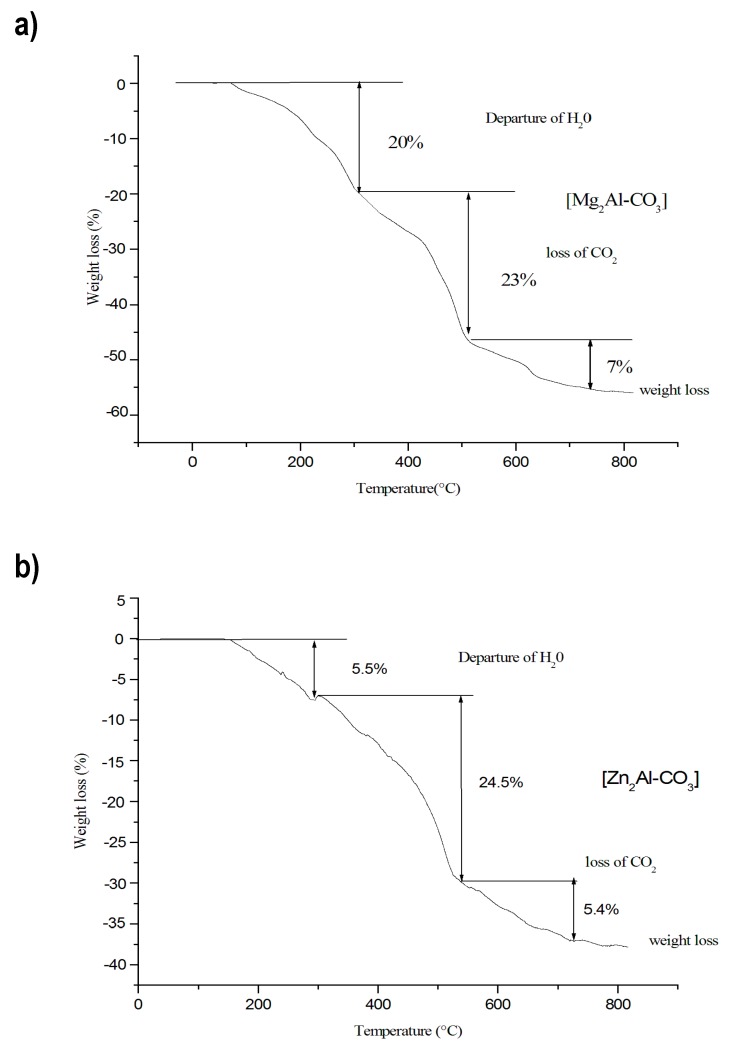
TGA of synthetic hydrotalcite. **a**) Mg_2_Al-CO_3_. **b**) Zn_2_Al-CO_3_.

**Figure 5 molecules-25-01263-f005:**
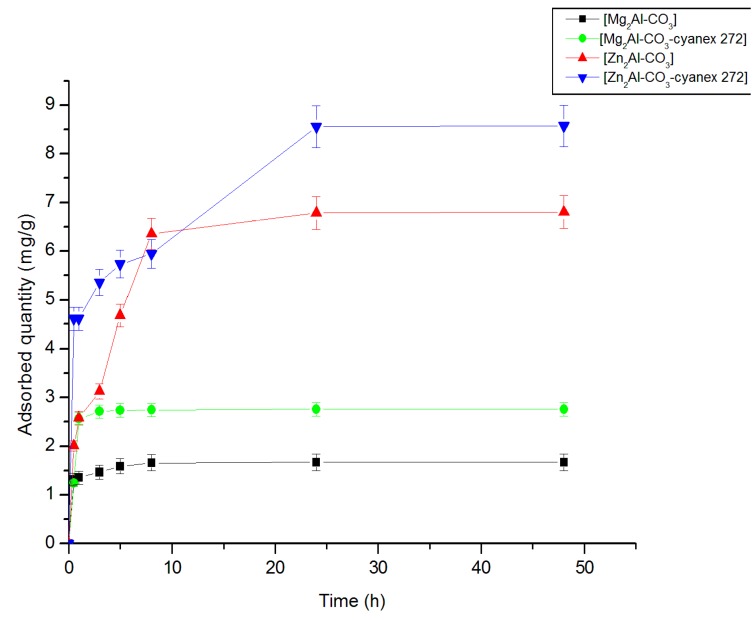
Effect of contact time on zinc removal (m = 0.1 g, pH_0_ = 1.00 ± 0.05, [Zn^+2^] = 100 mg.L^−1^, [Na^+^, H^+^-NO_3_^−^] = 0.1 M, T = 25 °C).

**Figure 6 molecules-25-01263-f006:**
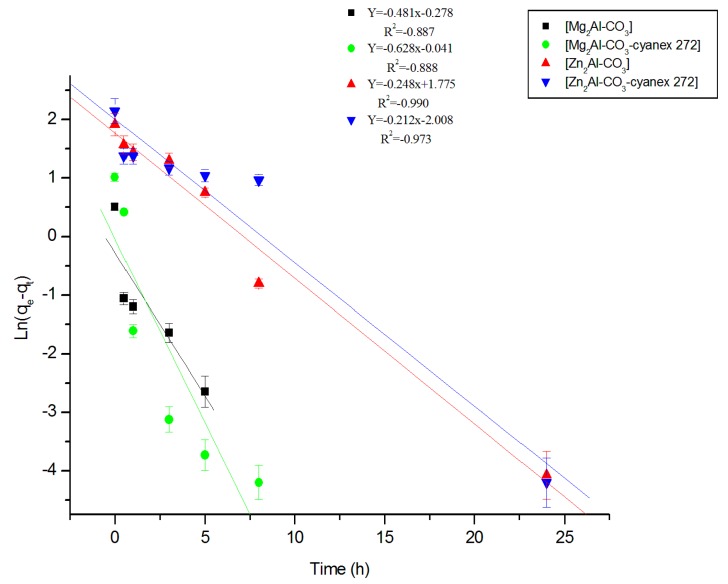
Kinetics of the pseudo-first order of Zn (II) extraction by the different materials.

**Figure 7 molecules-25-01263-f007:**
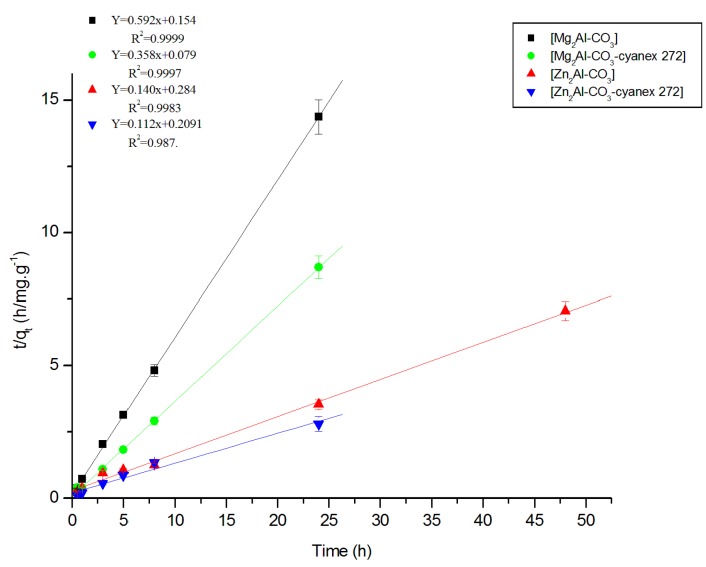
Kinetics of the pseudo-second order of Zn (II) extraction by the different materials.

**Figure 8 molecules-25-01263-f008:**
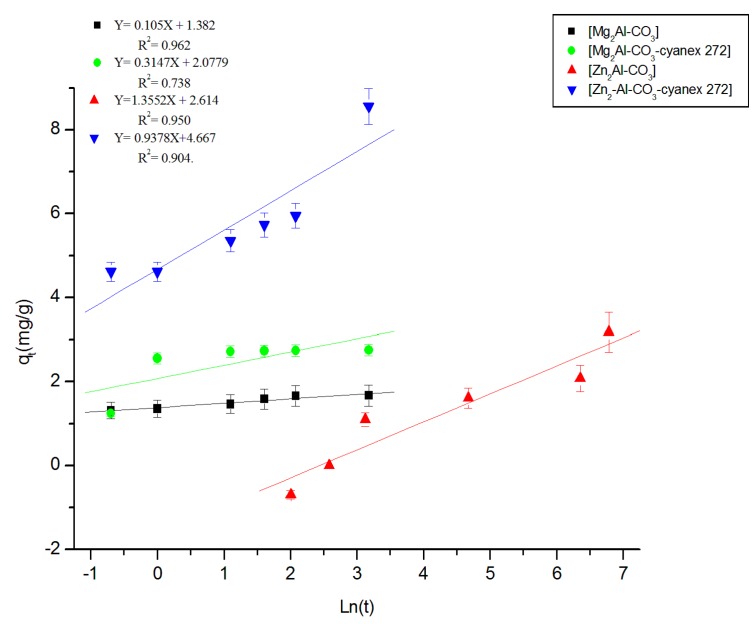
Elovich model of Zn (II) extraction by different materials.

**Figure 9 molecules-25-01263-f009:**
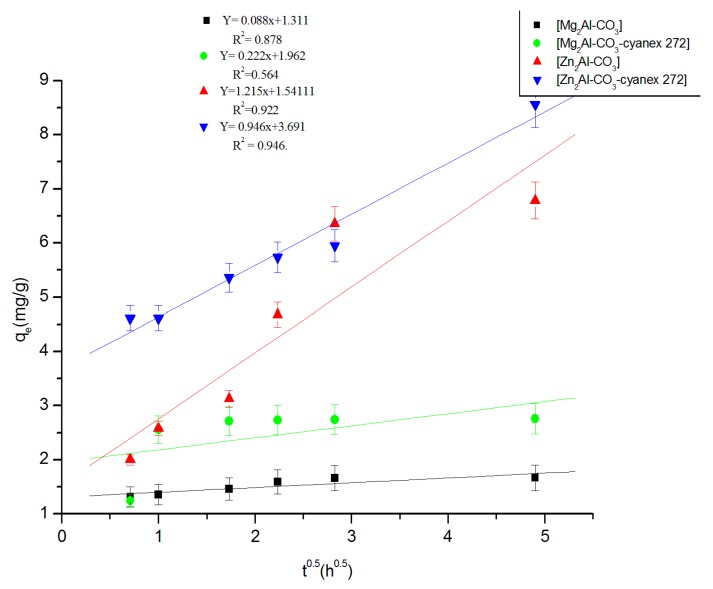
Kinetics of intraparticle diffusion of Zn (II) extraction.

**Figure 10 molecules-25-01263-f010:**
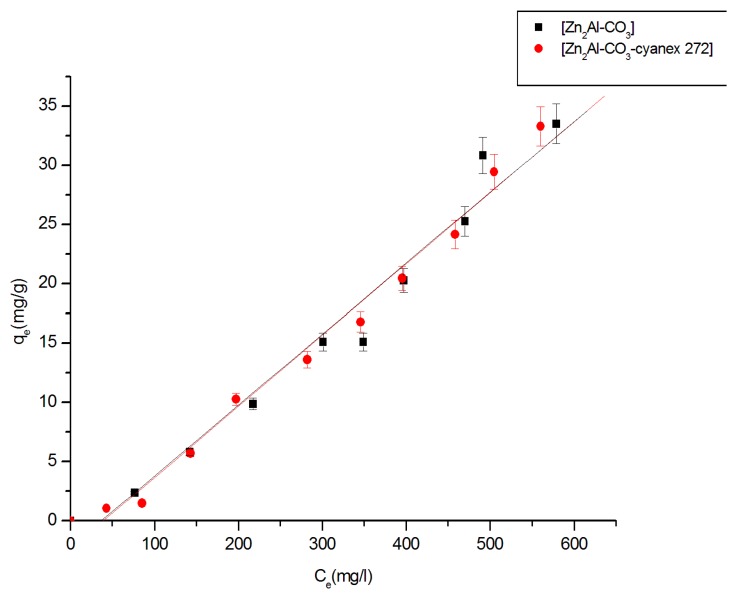
Isotherms of Zn(II) extraction (pH_0_ = 1.00 ± 0.05 [Zn^+2^] = 100 mg.L^−1^, [Na^+,^ H^+^-NO_3_^−^] = 0.1 M, T = 25 °C).

**Figure 11 molecules-25-01263-f011:**
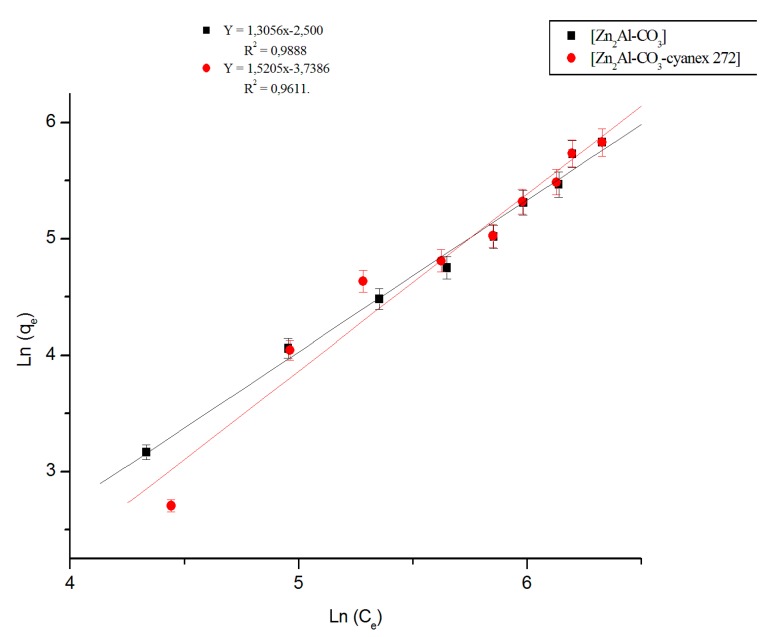
Freundlich isotherm of Zn(II) extraction.

**Figure 12 molecules-25-01263-f012:**
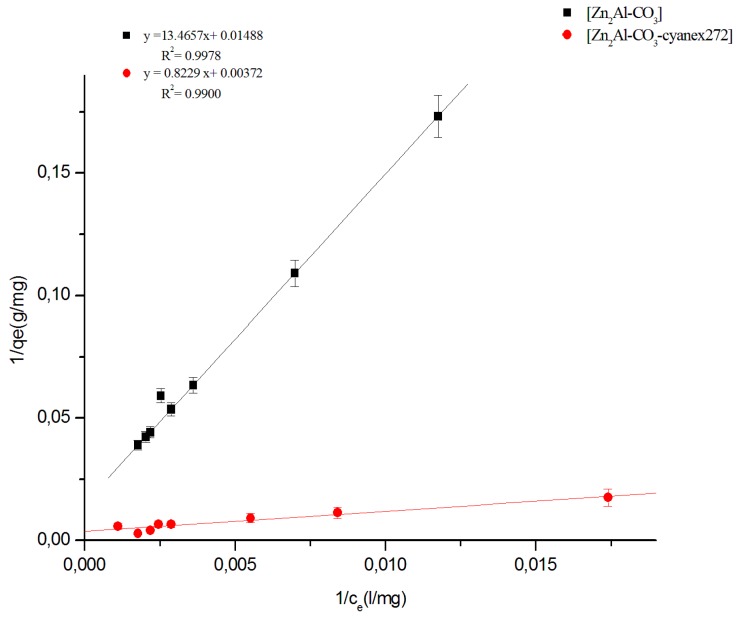
Langmuir isotherm of Zn(II) extraction.

**Table 1 molecules-25-01263-t001:** Unit cell parameters of Mg_2-_Al-CO_3_, Zn_2_Al-CO_3_, Zn_2-_Al-CO_3_-cyanex 272, and Mg_2_-Al-cyanex272 materials.

Material	*a *(Å)	*c *(Å)	d_003_ (Å)
Mg_2_Al-CO_3_	3.030	23.106	7.702
Zn_2_Al-CO_3_	3.043	23.517	7.839
Mg_2_Al-CO_3_-cyanex 272	3.004	48.000	15.99
Zn_2_Al-CO_3_-cyanex 272	3.055	49.950	16.65

**Table 2 molecules-25-01263-t002:** Kinetic parameters for Zn(II) extraction by the different materials.

Experimental		Pseudo-First Order			Pseudo-Second Order			Elovich			Intraparticular Diffusion		
Materials	q_exp _mg/g	k_1 _h^−1^	q_theo1 _mg/g	R^2^	k_2_g/mg.h	q_theo2 _mg/g	R^2^	Β g/mg	Α mg/g.h	R^2^	C mg/g	K_id _mg/g.h^1/2^	R^2^
Mg_2_Al-CO_3_	1.6	0.48	0.76	0.787	2.28	1.69	0.999	0.11	4.83 × 10^6^	0.962	1.31	0.09	0.878
Mg_2_Al-CO_3_-cyanex 272	2.8	0.63	0.96	0.789	1.62	2.79	0.997	0.32	2343.68	0.737	1.96	0.22	0.564
Zn_2_Al-CO_3_	6.2	0.25	5.90	0.970	0.07	7.14	0.998	1.36	5.07	0.951	1.54	1.22	0.922
Zn_2_Al-CO_3_-cyanex 272	8.8	0.21	7.45	0.946	3 × 10^−3^	8.92	0.987	0.94	154.58	0.904	3.69	0.95	0.986

**Table 3 molecules-25-01263-t003:** Langmuir and Freundlich constants and correlation coefficients.

Materials	Linear Regression		Langmuir Model			Freundlich Model		
	K_d_	R^2^	q_max_(mg/g)	K_l_ (L.g^−1^)	R^2^	n	K_F_ (mg^1−1/n ^.g^−1^.L^1/n^)	R^2^
Zn_2_Al-CO_3_	0.277	0.998	67.17	0.0011	0.9978	1.3056	2.0848	0.9887
Zn_2_Al-CO_3_- cyanex272	0.318	0.990	268.82	0.0045	0.9900	1.5205	3.3229	0.9611

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
