# Peer review of "Removal of Zinc from Aqueous Solutions Using Lamellar Double Hydroxide Materials Impregnated with Cyanex 272: Characterization and Sorption Studies"

_molecules, 2020, doi:10.3390/molecules25061263_

Round 1
Reviewer 1 Report
Manuscript Number: molecules-712747
Title: Removal of Zinc from Aqueous Solutions by Using Lamellar Double Hydroxide Materials Impregnated with Cyanex 272: Characterization and Sorption Studies
Article Type: article
In the manuscript the experimental research concerning synthesis of lamellar materials type layered double hydroxides (LDH) and their application in Zn (II) removal is presented. In the first part of manuscript a short theoretical introduction is made. Next the experimental section is detailed described. The manuscript is very well written. The data is clearly presented. I have just two small remarks for the authors:
The purity of materials used in the research should be given in the text. Error bars should be added in Fig. 5, 6, 7, 8, 9, 10, 11, 12. It is crucial for the analysis of obtained results. Moreover the information concerning the number of measurements should be given.I think that the manuscript is interesting and it can be published after minor revision.
Author Response
Hary Demey
Universitat Politecnica de Catalunya
Chemical Engineering Department
Av. Diagonal, 647.
08028 Barcelona, Spain
Email: hary.demey@upc.edu
Guest editors of Molecules:
Prof. Ming-Chung Wu
Dear Editors:
Please find enclosed the revised manuscript entitled “Removal of zinc using lamellar double hydroxide materials impregnated with cyanex 272: Characterization and sorption studies” to Molecules (MDPI Journal). We confirm that this work has not been published, nor it is currently under consideration for publication elsewhere.
We would like to thank the referees for the careful review of the work and their useful suggestions for improving it. A thorough revision has been conducted following all their relevant considerations. This manuscript describes a cheap cost manufacturing procedure of the lamellar double hydroxide (LDH)-based sorbents. It provides information about the impregnation with cyanex-272 extractant; which improves the selectivity toward zinc ions from aqueous effluents. The easy synthesis of the materials is promising for a further scaling-up of the process to demi-industrial size production.
In this work, the LDH materials were tested with zinc in order to approach the treatment to the real metallurgical effluents. This preliminary study is very useful for a large treatment with real wastewaters (currently on-going). The technique can be extrapolated to different species of metals, and is a doorway for a new generation of LDH-based sorbents. The influences of the main experimental parameters (i.e. pH, initial metal concentration, effect of contact time) were considered.
The characterization of the synthesized sorbents was performed with different techniques (e.g., XRD, TGA, FTIR) and the formation of the interlayers was demonstrated. As such, this paper could be of interest to a broad readership including those interested in chemical engineering, water treatment, materials science and biotechnology.
We hope you will agree this manuscript; however, we are prepared to modify the paper according to reviewers’ comments.
Sincerely,
PhD. Hary Demey
Reviewer #1:
In the manuscript the experimental research concerning synthesis of lamellar materials type layered double hydroxides (LDH) and their application in Zn (II) removal is presented. In the first part of manuscript a short theoretical introduction is made. Next the experimental section is detailed described. The manuscript is very well written. The data is clearly presented. I have just two small remarks for the authors:
- The purity of materials used in the research should be given in the text.
- Error bars should be added in Fig. 5, 6, 7, 8, 9, 10, 11, 12. It is crucial for the analysis of obtained results. Moreover the information concerning the number of measurements should be given. I think that the manuscript is interesting and it can be published after minor revision.
Authors:
We are very grateful to reviewer #1 for the valuable comments. The modifications were highlighted (in red color) in the whole text.
- Thank you for your recommendation. The purity of the materials has been indicated in the manuscript (page 2; section 2.1).
- All the Figures have been colored and the errors bars have been included (Figures 5 to 12). The number of measures has been indicated on page 4 (sections 2.5.1 and 2.5.2). The authors confirm that the experiments were performed in duplicate and the results are reproducible, the standard deviation was estimated in the order of + 2%.
Figure 2: XRD patterns of Mg2-Al-CO3, Zn2-Al-CO3, Zn2Al-CO3-cyanex 272 and Mg2Al-CO3-cyanex 272.
Figure 3: FTIR spectrum of the different materials and cyanex 272.
Figure 4: TGA of synthetic hydrotalcite. a) Mg2Al-CO3. b) Zn2Al-CO3.
Figure 5: Effect of contact time on zinc removal (m = 0.1 g, pH0 = 1.00±0.05, [Zn+2] = 100 mg.L-1, [Na+, H+-NO3-] = 0.1 M, T = 25 °C).
Figure 6: Kinetics of the pseudo-first order of Zn (II) extraction by the different materials.
Figure 7: Kinetics of the pseudo-second order of Zn (II) extraction by the different materials.
Figure 8: Elovich model of Zn (II) extraction by different materials.
Figure 9: Kinetics of intraparticle diffusion of Zn (II) extraction.
Figure 10: Isotherms of Zn(II) extraction (pH0 = 1.00 ± 0.05 [Zn+2] = 100 mg.L-1, [Na+, H+-NO3-] = 0.1 M, T = 25 °C).
Figure 11: Freundlich isotherm of Zn(II) extraction.
Figure 12: Langmuir isotherm of Zn(II) extraction.
Reviewer #2:
- Abstract. Synthesis details like those mentioned in the first 3 lines are not useful. Replace with a short description of the material.
- Introduction. The novelty of the study must be stated in the last paragraph.
- Section 2.5.1. It is mentioned "initial constant pH". pH can be adjusted initially but cannot remain constant.
- Clarify Section 2.5.2: What was the total sampling volume? Typically should be below 5%.
- Section 2.5.3: These are isotherms or equilibrium studies, not capacity measurements.
- Section 3.2: The models used are in the best case approximate. While they can be acceptable I strongly suggest to comment on the fallacies and refer to a recently paper discussing these issues: https://doi.org/10.1016/j.jhazmat.2018.12.023
Authors:
We are very grateful to reviewer #2 for the valuable comments.
- The abstract has been modified following the recommendations of the reviewer:
Abstract: Removal of heavy metals from wastewater is mandatory in order to avoid water pollution of the natural reservoirs. In the present study, layered double hydroxide materials LDH were evaluated for removal of zinc from aqueous solutions. Materials thus prepared were impregnated with cyanex 272 by dry method. These materials have been characterized by X-ray diffraction (XRD), Fourier transform infrared (FTIR) and thermal analysis. Batch shaking adsorption experiments were performed in order to examine contact time, and extraction capacity on removal process. Results show that the equilibrium time of Zn (II) extraction is about 4 h for Mg2Al-CO3 and Mg2Al-CO3-cyanex 272, 6 h for Zn2Al-CO3 and 24 h for Zn2Al-CO3-cyanex 272. The experimental equilibrium data were tested for Langmuir, and Freundlich isotherms models. Correlation coefficients indicate that experimental results are in a good agreement with Langmuir's model for zinc ions. Pseudo-first, second-order, Elovich and Intraparticular kinetic models were used for describing kinetic data. It was determined that removal of Zn2+ was well-fitted by second-order reaction kinetic. A maximum capacity of 280 mg/g was obtained by Zn2Al-CO3-cyanex 272
- The authors would like to thank the reviewer for the recommendation. In the last paragraph of the Introduction page 2 (Introduction), the authors have highlighted the relevance of this work:
Recently, some authors reported about LDHs containing different chelating agents [22-29], as well as the metal cations uptake achieved by these materials. Higher retention zinc capacities were obtained after impregnation of bentonite by cyanex 272 [28]. Referring to the literature, no studies on LDH functionalized with organophosphorus acids as a ligand for the trapping of metals have been cited. This work deals with the use of impregnated materials by cyanex 272 as an adsorbent of Zn (II). Extraction experiments at different conditions were performed. The sorption capacities, isotherms, kinetics and sorption mechanisms were determined and discussed in the next sections.
It is noteworthy that the impregnation of the LDH materials with cyanex 272 allows increasing four times the sorption capacity of the original sorbent, which is presented in the section of Results and Discussions (page 4). The manufacturing of potential sorbents with high removal uptake is crucial for achieving the industrial exploitation and commercial uses. The synthetic solutions of Zinc (II) have been evaluated in this work for understanding the involved mechanisms. Experimental evaluations are being performed currently with real metallurgical effluents (which will be the scope of a future work).
- The authors strongly agree with the reviewer. The pH of the solutions does not remain constant after contact with the sorbent materials during the experimental procedure. The authors have tried to explain in the section 2.5.1 that the initial pH of the solutions was systematically adjusted to a fixed initial pH 1, for the equilibrium studies. This section has been modified in order to clarify the experimental methodology.
The section 2.5.1 was modified:
The removal of Zn (II) from aqueous solutions was carried out through batch experiments at 25 °C. A fixed amount of sorbent (0.1 g) was mechanically mixed in polypropylene tubes with 10 ml of aqueous metal solution at a known initial metal concentration (and fixed initial pH 1), during 48 h of agitation time for achieving the equilibrium. Then, the solid phase was separated from the aqueous phase through centrifugation (at a speed of 8000 rpm); aliquots of five milliliters were withdrawn for analyzing with Atomic Absorption technique (AAS) on a Perkin-Elmer 2380 spectrophotometer. The experiments were performed in duplicate and the standard deviation was estimated in the order of + 2%.
Additionally, the authors would like to justify that initial pH 1 was chosen for avoiding the precipitation of zinc species. At this operating condition, the equilibrium pH of the metal solution after contact with the sorbent materials is pH 3-5 (Figure S1).
Figure S1. Variation in pH using the impregnated and non-impregnated LDH materials for zinc removal from aqueous solutions
The following paragraph was added to the section 3.2.1 (page 7):
The pH of the solution is a very important parameter to take into consideration in sorption processes. The sorbents materials were evaluated in this work at initial pH0: 1, in order to simulate the acid conditions of the real metallurgical effluents (in which zinc pollution is frequently present), and also for avoiding the metal precipitation due to the pH increases.
Figure S1 (supplementary materials section) reports the pH variation using the impregnated and non-impregnated LDH materials. As expected, after contact with the sorbents, the equilibrium pH of the effluents increases due to two main reasons: i) the competition between protons and metals ions for the active sites; ii) probably, an insufficient washing procedure (during the LDH manufacturing process), which could originate the release of trace amounts of NaOH on aqueous applications.
- The section 2.5.2 is related to the study of the contact time on sorption uptake. The experimental procedure was improved as follows:
The experimental protocol was performed following the procedure of the section 2.5.1. It means that several recipients containing 0.1 g of solid sorbent were prepared. Then it was added (in each one) 10 mL of the mother solution (at fixed initial pH and initial metal concentration). The recipients correspond to each sampling time (contact times ranging from 0 to 48 h); these were mixed with the sorbent and agitated at the same started time (t0). The agitation speed of the recipients was stopped according to the corresponding contact time, and the aqueous phase was immediately separated from the solid phase.
In this way, the variation of volume due to the aliquot sampling is avoided (the same contact volume is guaranteed for each sampling time). The experiments were performed twice for each sorbent materials and the standard deviation was + 5%. The pH was also monitored, and the obtained data were fitted with the pseudo-first order, pseudo-second order, Elovich and intraparticular diffusion models.
- Thank you to the reviewer for the valuable comments. The title of the section 2.5.3 (page 4) was modified as “Equilibrium studies”.
- The authors agree with the reviewer recommendation. The following comment was added on page 8 (section 3.2.2):
It is noteworthy that the fitting of the data does not mean that the principles of the models are verified, but it helps for understanding the involved mechanisms. Inglezakis et al. [45] pointed out that the sorption mechanism cannot be directly assigned by simply fitting the kinetics equations; the knowledge should be supported by combining the analytical surfaces techniques (e.g., FTIR, XRD, SEM, etc.).
Reviewer #3:
The article entitled: “Removal of Zinc from Aqueous Solutions by Using Lamellar Double Hydroxide Materials Impregnated with Cyanex 272: Characterization and Sorption Studies” presents the synthesis of two types of LDH materials Zn/Al and Mg/Al, their modification using Cyanex, and their application in the removal of Zn(II) from model aqueous solution.
- The title could be improved. I suggest to remove the word “by” or “using”. Also mistakes are present in affiliations, e.g. small letters.
- According to the Introduction – it should exhibit a constant structure in which the following elements should be added: Background (what is already known/what is a real problem to solve/etc) , Methods (should contain enough information to enable the reader to understand what was done), Results (should contain as much detail about the findings as the journal word count permits with numbers and values), and Conclusions (should contain the most important take-home message of the study, expressed in a few precisely worded sentences, some perspectives also).
- For me, the Abstract is a little bit chaotic, some elements are missing. Please improve it.
- In the Introduction section, Authors should present a problem of heavy metals occurrence more comprehensive, especially a problem of Zn (sources of occurrence, limitations, methods of removal, materials used as adsorbents etc.). Some literature may be helpful: https://doi.org/10.3390/min9080470
https://doi.org/10.1016/j.eti.2019.100464.
The goal emphasized in the Introduction shoul be more precise. Please indicate what has been already discovered/done and what is a novelty of your study.
- Please avoid using the word “extraction” instead of “adsorption”. These words are used interchangeably in the article.
- The section “Materials” is prepared well, however please correct the chemical formulas of some compounds you used.
- In section 2.4 Authors should provide a model and a producer of all devices. What is more, they missed to mention about TGA analysis and analyzer in this section. Please correct it.
- In the Results and Discussion, Authors should write “(…) is shown/presented in Figure xy” instead of”(…) is presented, Figure xy”. This sentence is repeated through the manuscript very often.
9.Coloured Figure 2 will make the picture more clear. Correct also the y axix. Could Authors provide an interpretation of obtained XRD results? especially what happened after cyanex modification?
- All figure captions should be consistent. Please correct the caption of figure 3 – do not use “different materials”.
- Figure presenting TGA analysis is of poor quality. I suggest to correct it. It will be also more valuable if Authors provide TGA analysis of the materials after Cyanex modification. The first paragraph in the 3.1.3 section should be transferred into the Materials and methods.
- There is a lack of holistic interpretation of the obtained results– the effect of Cyanex modification. Indeed, authors provide some references, however they did not refer to the other research, do not compare to the other results.
- The adsorption study revealed very unique adsorption isotherm. Why authors did not provide it for all 4 materials?
- The isotherm seems not be finished because there is no plateau. In many papers authors present the isotherms with some plateau points (when the adsorption capacity remains constant at high and higher C. I am not familiar with this type of isotherm, I will be satisfied if Authors show appropriate references where I can find similar results – and discuss it in the text.
Thus, the usage of Langmuir isotherm is very questionable and in my opinion it is a wrong direction. Langmuir isotherm is applicable for the situation where plateau is reached, that makes it clear that the maximal saturation has been achieved (monolayer capacity). In presented case the isotherm is not finished or the capacity will increase with the increase of the initial concentration suggesting the multilayer adsorption. Please refer to my doubts in your reply.
Kinetic investigations revealed that apart from different time needed to obtain an equilibrium state, all the materials revealed different maximal adsorption capacity. And I have the same comment as to the materials characterization - Authors provide some references, however they did not refer to the other research or do this not comprehensive. Taking into account that more investigations could be performed (both in materials characterization and in sorption studies) the discussion should be more detailed.
- Article should be checked by English language specialist
Authors:
We are very grateful to reviewer #3 for the valuable comments.
- The title was improved, the word “by” was deleted in the title, and the affiliations were corrected.
- The authors agree with the reviewer. The introduction was improved:
In natural environments, LDH minerals are significant to determine the uptake of heavy metal ions [19]. Divalent cations can be adsorbed by aluminum oxides and may form hydrotalcite-like minerals, such as Ni-Al LDH [20] and Zn-Al LDH [21]. This process can reduce heavy metal concentration in aquifers, and subsequent migration and bioavailability obviously. In the past few years, many reports on adsorption of various contaminants on LDHs have been published. The contaminants include oxyanions, monoatomic anions, organic compounds and gas. Only few studies have been focused on sorption of cations on LDHs. An important pollutant to be considered is zinc, which is highly present in industrial effluents (such as metallurgical and ceramic wastewaters). The introduction of waters contaminated with zinc into the ecosystems represents a serious environmental concern nowadays.
Recently, some authors reported about LDHs containing different chelating agents [22-29], as well as the metal cations uptake achieved by these materials. Higher retention zinc capacities were obtained after impregnation of bentonite by cyanex 272 [28]. Referring to the literature, no studies on LDH functionalized with organophosphorus acids as a ligand for the trapping of metals have been cited. This work deals with the use of impregnated materials by cyanex 272 as an adsorbent of Zn (II). Extraction experiments at different conditions were performed. The sorption capacities, isotherms, kinetics and sorption mechanisms were determined and discussed in the next sections.
The suggested references were added:
https://doi.org/10.3390/min9080470
https://doi.org/10.1016/j.eti.2019.100464
- The abstract has been modified following the recommendations of the reviewer:
Abstract: Removal of heavy metals from wastewater is mandatory in order to avoid water pollution of the natural reservoirs. In the present study, layered double hydroxide materials LDH were evaluated for removal of zinc from aqueous solutions. Materials thus prepared were impregnated with cyanex 272 by dry method. These materials have been characterized by X-ray diffraction (XRD), Fourier transform infrared (FTIR) and thermal analysis. Batch shaking adsorption experiments were performed in order to examine contact time, and extraction capacity on removal process. Results show that the equilibrium time of Zn (II) extraction is about 4 h for Mg2Al-CO3 and Mg2Al-CO3-cyanex 272, 6 h for Zn2Al-CO3 and 24 h for Zn2Al-CO3-cyanex 272. The experimental equilibrium data were tested for Langmuir, and Freundlich isotherms models. Correlation coefficients indicate that experimental results are in a good agreement with Langmuir's model for zinc ions. Pseudo-first, second-order, Elovich and Intraparticular kinetic models were used for describing kinetic data. It was determined that removal of Zn2+ was well-fitted by second-order reaction kinetic. A maximum capacity of 280 mg/g was obtained by Zn2Al-CO3-cyanex 272.
- The novelty of this work was highlighted in the last paragraph of the Introduction. The introduction has been improved as follows:
Recently, some authors reported about LDHs containing different chelating agents [22-29], as well as the metal cations uptake achieved by these materials. Higher retention zinc capacities were obtained after impregnation of bentonite by cyanex 272 [28]. Referring to the literature, no studies on LDH functionalized with organophosphorus acids as a ligand for the trapping of metals have been cited. This work deals with the use of impregnated materials by cyanex 272 as an adsorbent of Zn (II). Extraction experiments at different conditions were performed. The sorption capacities, isotherms, kinetics and sorption mechanisms were determined and discussed in the next sections.
The suggested references were added:
https://doi.org/10.3390/min9080470
https://doi.org/10.1016/j.eti.2019.100464
- The authors agree with the reviewer. In this work, both mechanisms are present in this work (using mostly the impregnated materials, in which the chelation of metals by the extraction mechanism with cyanex-272 is predominant). The authors used the word “Adsorption for the non-impregnated materials”, and the word “Extraction” for the impregnated materials with Cyanex 272. The authors are aware that both mechanisms are playing in the process, but in order to simplify and for making difference between both type of materials, these words were used in the text.
- The section of “Materials and methods” was corrected following the recommendations of the reviewer.
- The section 2.4 was improved, and the model and fabricant of all devices were added. The TGA analyses were also corrected.
- The authors thank the reviewer for this valuable recommendation. It was corrected in the whole text.
- The authors have colored all the plots, in order to improve the quality of the work, and the XRD discussion was improved. The introduction of cyanex-272 into the lamellar materials contributes o increase the interlayer space, in which the metal ions could be accommodated.
- The captions of all Figures were verified, and the caption of Figure 3 was corrected.
- The authors followed the reviewer recommendation. The first paragraph in the 3.1.3 section has been transferred into the Materials and methods.
The quality of Figure 3 (FTIR analyses) was improved. The resolution of Figure 4 (TGA analyses) was also improved; the thermal-gravimetric analyses of the materials after impregnation with Cyanex-272 were not presented, since the amount of produced gases has increased as a consequence of the decomposition of Cyanex-272 molecules. Thus, a better gas evacuation from the TGA equipment was required. A fraction of the gases is condensed into the system and clogs the evacuation pipes of the TGA equipment; however another non-condensed fraction could be valorized as syngas (CH4,CO, CO2 and H2).
- The effect of Cyanex-272 modification can be observed with the sorption isotherms, the impregnated materials are more efficient for Zn (II) removal than non-impregnated materials. This is very interesting from an industrial point of view, since the LDH materials could be used as solid supports of Cyanex-272 for zinc removal from aqueous effluents.
- The sorption isotherm studies have been performed with the best materials of this work, i.e. Zn2-Al-CO3 and Zn2-Al-CO3-Cyanex-272. The sorption capacities of Mg based materials are very low (in comparison with zinc-based materials), and from a point of view of industrial exploitation, the authors considered that isotherms studies for these sorbents are not relevant.
- The authors agree with the reviewer, however, according to Mishraa and Patel (2009) (https://doi:10.1016/j.jhazmat.2009.02.026), it was found that adsorption onto Bentonite is practically linear in solutions containing one metal specie, whereas in binary solution the linearity can only be observed for the lower concentrations. This may be due to the saturation of bentonite at higher concentration.
Thus, Nan Jing et al. (2018) (https://doi.org/10.1016/j.jhazmat.2018.02.049) reported that at low initial concentrations (less than 1000 mg/L), adsorption capacity of the synthesized hydroxy-apatite (HA) showed almost linear growth with the increase of initial concentrations until 2000 mg/L.
This could explain the linear trend of our sorbents. Probably the materials are not saturated at low concentrations levels (<600 mg/L), and the plateau is not reached. In the isotherms plots are usually observed two steps: i) A steeper one at low metal concentrations, which is due to gradient concentration between the core of the material and the solution; and ii) a second step in which the equilibrium is established and the saturation is completely reached.
The authors agree that Langmuir approach is not the most appropriated model to fit the experimental isotherms, but it could help the understanding of the involved mechanism.
Concerning the kinetic studies, a complete study was performed, and the evaluation of different kinetic models was carried out. This study provides useful information to readership interested in materials science and pollutants monitoring.
- The technical English was checked, and some minor mistakes were found and corrected.

Reviewer 2 Report
Abstract. Synthesis details like those mentioned in the first 3 lines are not useful. Replace with a short description of the material. Introduction. The novelty of the study must be stated in the last paragraph. Section 2.5.1. It is mentioned "initial constant pH". pH can be adjusted initially but cannot remain constant. Clarify Section 2.5.2. What was the total sampling volume? Typically should be below 5%. Section 2.5.3. These are isotherms or equilibrium studies, not capacity measurements. Section 3.2. The models used are in the best case approximate. While they can be acceptable i strongly suggest to comment on the fallacies and refer to a recently paper discussing these issues: https://doi.org/10.1016/j.jhazmat.2018.12.023Author Response
Hary Demey
Universitat Politecnica de Catalunya
Chemical Engineering Department
Av. Diagonal, 647.
08028 Barcelona, Spain
Email: hary.demey@upc.edu
Guest editors of Molecules:
Prof. Ming-Chung Wu
Dear Editors:
Please find enclosed the revised manuscript entitled “Removal of zinc using lamellar double hydroxide materials impregnated with cyanex 272: Characterization and sorption studies” to Molecules (MDPI Journal). We confirm that this work has not been published, nor it is currently under consideration for publication elsewhere.
We would like to thank the referees for the careful review of the work and their useful suggestions for improving it. A thorough revision has been conducted following all their relevant considerations. This manuscript describes a cheap cost manufacturing procedure of the lamellar double hydroxide (LDH)-based sorbents. It provides information about the impregnation with cyanex-272 extractant; which improves the selectivity toward zinc ions from aqueous effluents. The easy synthesis of the materials is promising for a further scaling-up of the process to demi-industrial size production.
In this work, the LDH materials were tested with zinc in order to approach the treatment to the real metallurgical effluents. This preliminary study is very useful for a large treatment with real wastewaters (currently on-going). The technique can be extrapolated to different species of metals, and is a doorway for a new generation of LDH-based sorbents. The influences of the main experimental parameters (i.e. pH, initial metal concentration, effect of contact time) were considered.
The characterization of the synthesized sorbents was performed with different techniques (e.g., XRD, TGA, FTIR) and the formation of the interlayers was demonstrated. As such, this paper could be of interest to a broad readership including those interested in chemical engineering, water treatment, materials science and biotechnology.
We hope you will agree this manuscript; however, we are prepared to modify the paper according to reviewers’ comments.
Sincerely,
PhD. Hary Demey
Reviewer #1:
In the manuscript the experimental research concerning synthesis of lamellar materials type layered double hydroxides (LDH) and their application in Zn (II) removal is presented. In the first part of manuscript a short theoretical introduction is made. Next the experimental section is detailed described. The manuscript is very well written. The data is clearly presented. I have just two small remarks for the authors:
- The purity of materials used in the research should be given in the text.
- Error bars should be added in Fig. 5, 6, 7, 8, 9, 10, 11, 12. It is crucial for the analysis of obtained results. Moreover the information concerning the number of measurements should be given. I think that the manuscript is interesting and it can be published after minor revision.
Authors:
We are very grateful to reviewer #1 for the valuable comments. The modifications were highlighted (in red color) in the whole text.
- Thank you for your recommendation. The purity of the materials has been indicated in the manuscript (page 2; section 2.1).
- All the Figures have been colored and the errors bars have been included (Figures 5 to 12). The number of measures has been indicated on page 4 (sections 2.5.1 and 2.5.2). The authors confirm that the experiments were performed in duplicate and the results are reproducible, the standard deviation was estimated in the order of + 2%.
Figure 2: XRD patterns of Mg2-Al-CO3, Zn2-Al-CO3, Zn2Al-CO3-cyanex 272 and Mg2Al-CO3-cyanex 272.
Figure 3: FTIR spectrum of the different materials and cyanex 272.
Figure 4: TGA of synthetic hydrotalcite. a) Mg2Al-CO3. b) Zn2Al-CO3.
Figure 5: Effect of contact time on zinc removal (m = 0.1 g, pH0 = 1.00±0.05, [Zn+2] = 100 mg.L-1, [Na+, H+-NO3-] = 0.1 M, T = 25 °C).
Figure 6: Kinetics of the pseudo-first order of Zn (II) extraction by the different materials.
Figure 7: Kinetics of the pseudo-second order of Zn (II) extraction by the different materials.
Figure 8: Elovich model of Zn (II) extraction by different materials.
Figure 9: Kinetics of intraparticle diffusion of Zn (II) extraction.
Figure 10: Isotherms of Zn(II) extraction (pH0 = 1.00 ± 0.05 [Zn+2] = 100 mg.L-1, [Na+, H+-NO3-] = 0.1 M, T = 25 °C).
Figure 11: Freundlich isotherm of Zn(II) extraction.
Figure 12: Langmuir isotherm of Zn(II) extraction.
Reviewer #2:
- Abstract. Synthesis details like those mentioned in the first 3 lines are not useful. Replace with a short description of the material.
- Introduction. The novelty of the study must be stated in the last paragraph.
- Section 2.5.1. It is mentioned "initial constant pH". pH can be adjusted initially but cannot remain constant.
- Clarify Section 2.5.2: What was the total sampling volume? Typically should be below 5%.
- Section 2.5.3: These are isotherms or equilibrium studies, not capacity measurements.
- Section 3.2: The models used are in the best case approximate. While they can be acceptable I strongly suggest to comment on the fallacies and refer to a recently paper discussing these issues: https://doi.org/10.1016/j.jhazmat.2018.12.023
Authors:
We are very grateful to reviewer #2 for the valuable comments.
- The abstract has been modified following the recommendations of the reviewer:
Abstract: Removal of heavy metals from wastewater is mandatory in order to avoid water pollution of the natural reservoirs. In the present study, layered double hydroxide materials LDH were evaluated for removal of zinc from aqueous solutions. Materials thus prepared were impregnated with cyanex 272 by dry method. These materials have been characterized by X-ray diffraction (XRD), Fourier transform infrared (FTIR) and thermal analysis. Batch shaking adsorption experiments were performed in order to examine contact time, and extraction capacity on removal process. Results show that the equilibrium time of Zn (II) extraction is about 4 h for Mg2Al-CO3 and Mg2Al-CO3-cyanex 272, 6 h for Zn2Al-CO3 and 24 h for Zn2Al-CO3-cyanex 272. The experimental equilibrium data were tested for Langmuir, and Freundlich isotherms models. Correlation coefficients indicate that experimental results are in a good agreement with Langmuir's model for zinc ions. Pseudo-first, second-order, Elovich and Intraparticular kinetic models were used for describing kinetic data. It was determined that removal of Zn2+ was well-fitted by second-order reaction kinetic. A maximum capacity of 280 mg/g was obtained by Zn2Al-CO3-cyanex 272
- The authors would like to thank the reviewer for the recommendation. In the last paragraph of the Introduction page 2 (Introduction), the authors have highlighted the relevance of this work:
Recently, some authors reported about LDHs containing different chelating agents [22-29], as well as the metal cations uptake achieved by these materials. Higher retention zinc capacities were obtained after impregnation of bentonite by cyanex 272 [28]. Referring to the literature, no studies on LDH functionalized with organophosphorus acids as a ligand for the trapping of metals have been cited. This work deals with the use of impregnated materials by cyanex 272 as an adsorbent of Zn (II). Extraction experiments at different conditions were performed. The sorption capacities, isotherms, kinetics and sorption mechanisms were determined and discussed in the next sections.
It is noteworthy that the impregnation of the LDH materials with cyanex 272 allows increasing four times the sorption capacity of the original sorbent, which is presented in the section of Results and Discussions (page 4). The manufacturing of potential sorbents with high removal uptake is crucial for achieving the industrial exploitation and commercial uses. The synthetic solutions of Zinc (II) have been evaluated in this work for understanding the involved mechanisms. Experimental evaluations are being performed currently with real metallurgical effluents (which will be the scope of a future work).
- The authors strongly agree with the reviewer. The pH of the solutions does not remain constant after contact with the sorbent materials during the experimental procedure. The authors have tried to explain in the section 2.5.1 that the initial pH of the solutions was systematically adjusted to a fixed initial pH 1, for the equilibrium studies. This section has been modified in order to clarify the experimental methodology.
The section 2.5.1 was modified:
The removal of Zn (II) from aqueous solutions was carried out through batch experiments at 25 °C. A fixed amount of sorbent (0.1 g) was mechanically mixed in polypropylene tubes with 10 ml of aqueous metal solution at a known initial metal concentration (and fixed initial pH 1), during 48 h of agitation time for achieving the equilibrium. Then, the solid phase was separated from the aqueous phase through centrifugation (at a speed of 8000 rpm); aliquots of five milliliters were withdrawn for analyzing with Atomic Absorption technique (AAS) on a Perkin-Elmer 2380 spectrophotometer. The experiments were performed in duplicate and the standard deviation was estimated in the order of + 2%.
Additionally, the authors would like to justify that initial pH 1 was chosen for avoiding the precipitation of zinc species. At this operating condition, the equilibrium pH of the metal solution after contact with the sorbent materials is pH 3-5 (Figure S1).
Figure S1. Variation in pH using the impregnated and non-impregnated LDH materials for zinc removal from aqueous solutions
The following paragraph was added to the section 3.2.1 (page 7):
The pH of the solution is a very important parameter to take into consideration in sorption processes. The sorbents materials were evaluated in this work at initial pH0: 1, in order to simulate the acid conditions of the real metallurgical effluents (in which zinc pollution is frequently present), and also for avoiding the metal precipitation due to the pH increases.
Figure S1 (supplementary materials section) reports the pH variation using the impregnated and non-impregnated LDH materials. As expected, after contact with the sorbents, the equilibrium pH of the effluents increases due to two main reasons: i) the competition between protons and metals ions for the active sites; ii) probably, an insufficient washing procedure (during the LDH manufacturing process), which could originate the release of trace amounts of NaOH on aqueous applications.
- The section 2.5.2 is related to the study of the contact time on sorption uptake. The experimental procedure was improved as follows:
The experimental protocol was performed following the procedure of the section 2.5.1. It means that several recipients containing 0.1 g of solid sorbent were prepared. Then it was added (in each one) 10 mL of the mother solution (at fixed initial pH and initial metal concentration). The recipients correspond to each sampling time (contact times ranging from 0 to 48 h); these were mixed with the sorbent and agitated at the same started time (t0). The agitation speed of the recipients was stopped according to the corresponding contact time, and the aqueous phase was immediately separated from the solid phase.
In this way, the variation of volume due to the aliquot sampling is avoided (the same contact volume is guaranteed for each sampling time). The experiments were performed twice for each sorbent materials and the standard deviation was + 5%. The pH was also monitored, and the obtained data were fitted with the pseudo-first order, pseudo-second order, Elovich and intraparticular diffusion models.
- Thank you to the reviewer for the valuable comments. The title of the section 2.5.3 (page 4) was modified as “Equilibrium studies”.
- The authors agree with the reviewer recommendation. The following comment was added on page 8 (section 3.2.2):
It is noteworthy that the fitting of the data does not mean that the principles of the models are verified, but it helps for understanding the involved mechanisms. Inglezakis et al. [45] pointed out that the sorption mechanism cannot be directly assigned by simply fitting the kinetics equations; the knowledge should be supported by combining the analytical surfaces techniques (e.g., FTIR, XRD, SEM, etc.).
Reviewer #3:
The article entitled: “Removal of Zinc from Aqueous Solutions by Using Lamellar Double Hydroxide Materials Impregnated with Cyanex 272: Characterization and Sorption Studies” presents the synthesis of two types of LDH materials Zn/Al and Mg/Al, their modification using Cyanex, and their application in the removal of Zn(II) from model aqueous solution.
- The title could be improved. I suggest to remove the word “by” or “using”. Also mistakes are present in affiliations, e.g. small letters.
- According to the Introduction – it should exhibit a constant structure in which the following elements should be added: Background (what is already known/what is a real problem to solve/etc) , Methods (should contain enough information to enable the reader to understand what was done), Results (should contain as much detail about the findings as the journal word count permits with numbers and values), and Conclusions (should contain the most important take-home message of the study, expressed in a few precisely worded sentences, some perspectives also).
- For me, the Abstract is a little bit chaotic, some elements are missing. Please improve it.
- In the Introduction section, Authors should present a problem of heavy metals occurrence more comprehensive, especially a problem of Zn (sources of occurrence, limitations, methods of removal, materials used as adsorbents etc.). Some literature may be helpful: https://doi.org/10.3390/min9080470
https://doi.org/10.1016/j.eti.2019.100464.
The goal emphasized in the Introduction shoul be more precise. Please indicate what has been already discovered/done and what is a novelty of your study.
- Please avoid using the word “extraction” instead of “adsorption”. These words are used interchangeably in the article.
- The section “Materials” is prepared well, however please correct the chemical formulas of some compounds you used.
- In section 2.4 Authors should provide a model and a producer of all devices. What is more, they missed to mention about TGA analysis and analyzer in this section. Please correct it.
- In the Results and Discussion, Authors should write “(…) is shown/presented in Figure xy” instead of”(…) is presented, Figure xy”. This sentence is repeated through the manuscript very often.
9.Coloured Figure 2 will make the picture more clear. Correct also the y axix. Could Authors provide an interpretation of obtained XRD results? especially what happened after cyanex modification?
- All figure captions should be consistent. Please correct the caption of figure 3 – do not use “different materials”.
- Figure presenting TGA analysis is of poor quality. I suggest to correct it. It will be also more valuable if Authors provide TGA analysis of the materials after Cyanex modification. The first paragraph in the 3.1.3 section should be transferred into the Materials and methods.
- There is a lack of holistic interpretation of the obtained results– the effect of Cyanex modification. Indeed, authors provide some references, however they did not refer to the other research, do not compare to the other results.
- The adsorption study revealed very unique adsorption isotherm. Why authors did not provide it for all 4 materials?
- The isotherm seems not be finished because there is no plateau. In many papers authors present the isotherms with some plateau points (when the adsorption capacity remains constant at high and higher C. I am not familiar with this type of isotherm, I will be satisfied if Authors show appropriate references where I can find similar results – and discuss it in the text.
Thus, the usage of Langmuir isotherm is very questionable and in my opinion it is a wrong direction. Langmuir isotherm is applicable for the situation where plateau is reached, that makes it clear that the maximal saturation has been achieved (monolayer capacity). In presented case the isotherm is not finished or the capacity will increase with the increase of the initial concentration suggesting the multilayer adsorption. Please refer to my doubts in your reply.
Kinetic investigations revealed that apart from different time needed to obtain an equilibrium state, all the materials revealed different maximal adsorption capacity. And I have the same comment as to the materials characterization - Authors provide some references, however they did not refer to the other research or do this not comprehensive. Taking into account that more investigations could be performed (both in materials characterization and in sorption studies) the discussion should be more detailed.
- Article should be checked by English language specialist
Authors:
We are very grateful to reviewer #3 for the valuable comments.
- The title was improved, the word “by” was deleted in the title, and the affiliations were corrected.
- The authors agree with the reviewer. The introduction was improved:
In natural environments, LDH minerals are significant to determine the uptake of heavy metal ions [19]. Divalent cations can be adsorbed by aluminum oxides and may form hydrotalcite-like minerals, such as Ni-Al LDH [20] and Zn-Al LDH [21]. This process can reduce heavy metal concentration in aquifers, and subsequent migration and bioavailability obviously. In the past few years, many reports on adsorption of various contaminants on LDHs have been published. The contaminants include oxyanions, monoatomic anions, organic compounds and gas. Only few studies have been focused on sorption of cations on LDHs. An important pollutant to be considered is zinc, which is highly present in industrial effluents (such as metallurgical and ceramic wastewaters). The introduction of waters contaminated with zinc into the ecosystems represents a serious environmental concern nowadays.
Recently, some authors reported about LDHs containing different chelating agents [22-29], as well as the metal cations uptake achieved by these materials. Higher retention zinc capacities were obtained after impregnation of bentonite by cyanex 272 [28]. Referring to the literature, no studies on LDH functionalized with organophosphorus acids as a ligand for the trapping of metals have been cited. This work deals with the use of impregnated materials by cyanex 272 as an adsorbent of Zn (II). Extraction experiments at different conditions were performed. The sorption capacities, isotherms, kinetics and sorption mechanisms were determined and discussed in the next sections.
The suggested references were added:
https://doi.org/10.3390/min9080470
https://doi.org/10.1016/j.eti.2019.100464
- The abstract has been modified following the recommendations of the reviewer:
Abstract: Removal of heavy metals from wastewater is mandatory in order to avoid water pollution of the natural reservoirs. In the present study, layered double hydroxide materials LDH were evaluated for removal of zinc from aqueous solutions. Materials thus prepared were impregnated with cyanex 272 by dry method. These materials have been characterized by X-ray diffraction (XRD), Fourier transform infrared (FTIR) and thermal analysis. Batch shaking adsorption experiments were performed in order to examine contact time, and extraction capacity on removal process. Results show that the equilibrium time of Zn (II) extraction is about 4 h for Mg2Al-CO3 and Mg2Al-CO3-cyanex 272, 6 h for Zn2Al-CO3 and 24 h for Zn2Al-CO3-cyanex 272. The experimental equilibrium data were tested for Langmuir, and Freundlich isotherms models. Correlation coefficients indicate that experimental results are in a good agreement with Langmuir's model for zinc ions. Pseudo-first, second-order, Elovich and Intraparticular kinetic models were used for describing kinetic data. It was determined that removal of Zn2+ was well-fitted by second-order reaction kinetic. A maximum capacity of 280 mg/g was obtained by Zn2Al-CO3-cyanex 272.
- The novelty of this work was highlighted in the last paragraph of the Introduction. The introduction has been improved as follows:
Recently, some authors reported about LDHs containing different chelating agents [22-29], as well as the metal cations uptake achieved by these materials. Higher retention zinc capacities were obtained after impregnation of bentonite by cyanex 272 [28]. Referring to the literature, no studies on LDH functionalized with organophosphorus acids as a ligand for the trapping of metals have been cited. This work deals with the use of impregnated materials by cyanex 272 as an adsorbent of Zn (II). Extraction experiments at different conditions were performed. The sorption capacities, isotherms, kinetics and sorption mechanisms were determined and discussed in the next sections.
The suggested references were added:
https://doi.org/10.3390/min9080470
https://doi.org/10.1016/j.eti.2019.100464
- The authors agree with the reviewer. In this work, both mechanisms are present in this work (using mostly the impregnated materials, in which the chelation of metals by the extraction mechanism with cyanex-272 is predominant). The authors used the word “Adsorption for the non-impregnated materials”, and the word “Extraction” for the impregnated materials with Cyanex 272. The authors are aware that both mechanisms are playing in the process, but in order to simplify and for making difference between both type of materials, these words were used in the text.
- The section of “Materials and methods” was corrected following the recommendations of the reviewer.
- The section 2.4 was improved, and the model and fabricant of all devices were added. The TGA analyses were also corrected.
- The authors thank the reviewer for this valuable recommendation. It was corrected in the whole text.
- The authors have colored all the plots, in order to improve the quality of the work, and the XRD discussion was improved. The introduction of cyanex-272 into the lamellar materials contributes o increase the interlayer space, in which the metal ions could be accommodated.
- The captions of all Figures were verified, and the caption of Figure 3 was corrected.
- The authors followed the reviewer recommendation. The first paragraph in the 3.1.3 section has been transferred into the Materials and methods.
The quality of Figure 3 (FTIR analyses) was improved. The resolution of Figure 4 (TGA analyses) was also improved; the thermal-gravimetric analyses of the materials after impregnation with Cyanex-272 were not presented, since the amount of produced gases has increased as a consequence of the decomposition of Cyanex-272 molecules. Thus, a better gas evacuation from the TGA equipment was required. A fraction of the gases is condensed into the system and clogs the evacuation pipes of the TGA equipment; however another non-condensed fraction could be valorized as syngas (CH4,CO, CO2 and H2).
- The effect of Cyanex-272 modification can be observed with the sorption isotherms, the impregnated materials are more efficient for Zn (II) removal than non-impregnated materials. This is very interesting from an industrial point of view, since the LDH materials could be used as solid supports of Cyanex-272 for zinc removal from aqueous effluents.
- The sorption isotherm studies have been performed with the best materials of this work, i.e. Zn2-Al-CO3 and Zn2-Al-CO3-Cyanex-272. The sorption capacities of Mg based materials are very low (in comparison with zinc-based materials), and from a point of view of industrial exploitation, the authors considered that isotherms studies for these sorbents are not relevant.
- The authors agree with the reviewer, however, according to Mishraa and Patel (2009) (https://doi:10.1016/j.jhazmat.2009.02.026), it was found that adsorption onto Bentonite is practically linear in solutions containing one metal specie, whereas in binary solution the linearity can only be observed for the lower concentrations. This may be due to the saturation of bentonite at higher concentration.
Thus, Nan Jing et al. (2018) (https://doi.org/10.1016/j.jhazmat.2018.02.049) reported that at low initial concentrations (less than 1000 mg/L), adsorption capacity of the synthesized hydroxy-apatite (HA) showed almost linear growth with the increase of initial concentrations until 2000 mg/L.
This could explain the linear trend of our sorbents. Probably the materials are not saturated at low concentrations levels (<600 mg/L), and the plateau is not reached. In the isotherms plots are usually observed two steps: i) A steeper one at low metal concentrations, which is due to gradient concentration between the core of the material and the solution; and ii) a second step in which the equilibrium is established and the saturation is completely reached.
The authors agree that Langmuir approach is not the most appropriated model to fit the experimental isotherms, but it could help the understanding of the involved mechanism.
Concerning the kinetic studies, a complete study was performed, and the evaluation of different kinetic models was carried out. This study provides useful information to readership interested in materials science and pollutants monitoring.
- The technical English was checked, and some minor mistakes were found and corrected.

Reviewer 3 Report
The article entitled: „Removal of Zinc from Aqueous Solutions by Using Lamellar Double Hydroxide Materials Impregnated with Cyanex 272: Characterization and Sorption Studies” presents the synthesis of two types of LDH materials Zn/Al and Mg/Al, their modification using Cyanex, and their application in the removal of Zn(II) from model aqueous solution.
The title could be improved. I suggest to remove the word „by” or „using”. Also mistakes are present in afiliations, e.g. small letters.
According to the Introduction – it should exhibit a constant structure in which the following elements should be added: Background (what is already known/what is a real problem to solve/etc) , Methods (should contain enough information to enable the reader to understand what was done), Results (should contain as much detail about the findings as the journal word count permits with numbers and values), and Conclusions (should contain the most important take-home message of the study, expressed in a few precisely worded sentences, some perspectives also). For me, the Abstract is a little bit chaotic, some elements are missing. Please improve it.
In the Introduction section, Authors should present a problem of heavy metals occurrence more comprehensive, especially a problem of Zn (sources of occurrence, limitations, methods of removal, materials used as adsorbents etc.). Some literature may be helpful:
https://doi.org/10.3390/min9080470 https://doi.org/10.1016/j.eti.2019.100464.The goal emhasised in the Introduction shoul be more precise. Please indicate what has been already discovered/done and what is a novelty of your study. Please avoid using the word „extraction” instead of „adsorption”. These words are used interchangeably in the article.
The section „Materials” is prepared well, however please correct the chemical formulas of some compounds you used. In section 2.4 Authors should provide a model and a producer of all devices. What is more, they missed to mention about TGA analysis and anlyser in this section. Please correct it.
In the Results and Discussion, Authors should write „(…) is shown/presented in Figure xy” instead of”(…) is presented, Figure xy”. This sentence is repeated throuh the manuscript very often. Coloured Figure 2 will make the picure more clear. Correct also the y axix. Could Authors provide an interpretation of obtained XRD results? espetially what happened after cyanex modification ?
All figure captions should be consistent. Please correct the caption of figure 3 – do not use „different materials”. Figure presenting TGA analysis is of poor quality. I suggest to correct it. It will be also more valuable if Autors provide TGA analysis of the materials after Cyanex modification. The first paragraph in the 3.1.3 section should be transferred int the Materials and methods.
There is a lack of holistic interpretation of the obtained results – the effect of Cyanex modification. Indeed, Authors provide some references, hovever they did not reffer to the other research, do not compare to the other results.
The adsorption study revealed very unique adsorption isotherm. Why authors did not provide it for all 4 materials? The isotherm seems not be finished because there is no plateau. In many papers authors present the isotherms with some plateau points (when the adsorption capacity remains constant at high and higher C0 . I am not familiar with this type of isotherm, I will be satisfied if Authors show appropriate references where I can find similar results – and discuss it in the text.
Thus, the usage of Langmuir isotherm is very questionable and in my opinion it is a wrong direction. Langmuir isotherm is applicable for the situation where plateau is reached, that makes it clear that the maximal saturation has been achieved (monolayer capacity). In presented case the isotherm is not finished or the capacity will increase with the increase of the initial concentration suggesting the multilayer adsorption. Please reffer to my doubts in your reply.
Kinetic investigations revealed that apart from differetn time needed to obtain an equilibrium state, all the materials revealed different maximal adsorption capacity.
And I have the same comment as to the materials characterisation - Authors provide some references, hovever they did not reffer to the other research or do this not comprehensive. Taking inta account that more investigations could be performed (both in materials characterisation and in sorption studies) the discussion should be more detailed.
Artilce should be checked by English language specialist.
Author Response

(The authors gave the same response as above.)

Round 2
Reviewer 2 Report
The paper can be accepted. Language editing is needed.